# Discovery of a novel symbiotic lineage associated with a hematophagous leech from the genus *Haementeria*

Víctor Manuel Sosa-Jiménez,[1,2] Sebastian Kvist,[3,4] Alejandro Manzano-Marín,[5] Alejandro Oceguera-Figueroa[2]

**ABSTRACT** Similarly to other strict blood feeders, leeches from the *Haementeria* genus (Hirudinida: Glossiphoniidae) have established a symbiotic association with bacteria harbored intracellularly in esophageal bacteriomes. Previous genome sequence analyses of these endosymbionts revealed co-divergence with their hosts, a strong genome reduction, and a simplified metabolism largely dedicated to the production of B vitamins, which are nutrients lacking from a blood diet. 'Candidatus Providencia siddallii' has been identified as the obligate nutritional endosymbiont of a monophyletic clade of Mexican and South American *Haementeria* spp. However, the *Haementeria* genus includes a sister clade of congeners from Central and South America, where the presence or absence of the aforementioned symbiont taxon remains unknown. In this work, we report on a novel bacterial endosymbiont found in a representative from this *Haementeria* clade. We found that this symbiont lineage has evolved from within the *Pluralibacter* genus, known mainly from clinical but also environmental strains. Similarly to *Ca. Providencia siddallii*, the *Haementeria*-associated *Pluralibacter* symbiont displays clear signs of genome reduction, accompanied by an A+T-biased sequence composition. Genomic analysis of its metabolic potential revealed a retention of pathways related to B vitamin biosynthesis, supporting its role as a nutritional endosymbiont. Finally, comparative genomics of both *Haementeria* symbiont lineages suggests that an ancient *Providencia* symbiont was likely replaced by the novel *Pluralibacter* one, thus constituting the first reported case of nutritional symbiont replacement in a leech without morphological changes in the bacteriome.

**IMPORTANCE** Obligate symbiotic associations with a nutritional base have likely evolved more than once in strict blood-feeding leeches. Unlike those symbioses found in hematophagous arthropods, the nature, identity, and evolutionary history of these remains poorly studied. In this work, we further explored obligate nutritional associations between *Haementeria* leeches and their microbial symbionts, which led to the unexpected discovery of a novel symbiosis with a member of the *Pluralibacter* genus. When compared to *Providencia siddallii*, an obligate nutritional symbiont of other *Haementeria* leeches, this novel bacterial symbiont shows convergent retention of the metabolic pathways involved in B vitamin biosynthesis. Moreover, the genomic characteristics of this *Pluralibacter* symbiont suggest a more recent association than that of *Pr. siddallii* and *Haementeria*. We conclude that the once-thought stable associations between blood-feeding Glossiphoniidae and their symbionts (i.e., one bacteriome structure, one symbiont lineage) can break down, mirroring symbiont turnover observed in various arthropod lineages.

**KEYWORDS** Hirudinida, leech symbiont, symbiont replacement, blood feeder, B vitamin, *Pluralibacter*

Address correspondence to Alejandro Manzano-Marín, alejandro.manzano.marin@univie.ac.at.

Alejandro Manzano-Marín and Alejandro Oceguera-Figueroa contributed equally to this article.

The authors declare no conflict of interest.

See the funding table on p. 9.

Obligate symbiosis is a widespread phenomenon across a variety of animals, particularly within those with a nutrient-limited diet [e.g., plant sap or blood (1–4)]. Once these obligate associates establish a stable vertical transmission, their genomes undergo radical changes, which include loss of dispensable genes/pathways, an invasion by and subsequent loss of mobile elements, and a shift in nucleotide composition (i.e., becoming more commonly A+T-biased) (5–7). This genome reduction syndrome can eventually lead to the impairment of essential functions of the symbiotic bacteria due to pseudogenization or gene loss affecting key pathways to its own maintenance or that of its host. This impairment can lead to one of three outcomes: breakdown of the symbiosis, co-obligate symbiont acquisition (i.e., a novel symbiont rescuing losses from the older one), or symbiont replacement (8, 9).

Symbiont replacement has been observed in many clades of insects (1, 10, 11), including blood-feeding arthropods (12). The novel symbiont can colonize the niche of its predecessor (13), which is evidenced by the conservation of the bacteriome location and structure, and can also occupy a different tissue (1, 14). If the degenerated symbiont is replaced by another microbe with a more competent genome, the replacement can result in novel metabolic capabilities for the host–symbiont system. Thus, symbiont replacement, similarly to symbiont acquisition, has been proposed as a source of ecological innovation (15). If the novel symbiotic bacterium establishes a stable association with its host throughout evolutionary time, it may follow a similar genome degeneration path as the former (now extinct) symbiont that can eventually lead to any of the aforementioned outcomes.

One of the best-studied groups of animals with a nutrient-restricted diet is sap feeders, which face a strong dietary deficiency of essential amino acids and B vitamins (16, 17). A group that has historically received less attention is hematophagous animals, which are confronted with a strong B vitamin deficiency. Not surprisingly, similarly to sap feeders, these obligate blood-feeding organisms house obligate symbionts that are able to compensate for their nutrient-deficient diet (18–20). Often, these microorganisms have been found to inhabit intracellularly in specialized host cells referred to as bacteriocytes, which together make up so-called bacteriomes (21–24), and have evolved small genomes with reduced gene repertoires that convergently retain B vitamin biosynthetic pathways.

Leeches are a monophyletic group of annelids, which are both famous and infamous for their blood-feeding habit (25). Species in the Glossiphoniidae family are proboscis-bearing leeches, whose blood-feeding members have evolved different bacteriome morphologies to host endosymbiotic bacteria, namely, belonging to the alpha- and gammaproteobacteria (21, 22, 24, 26). In particular, species of the *Haementeria* genus can be distinguished for having two pairs of globular sacs attached by thin ducts to the esophagus (21, 27). These bacteriomes harbor the gammaproteobacterial symbiont '*Candidatus* Providencia siddallii' (hereafter *Pr. siddallii*), which has been shown to hold small A+T-biased genome-preserving complete pathways for B vitamin biosynthesis (28). These characteristics, along with congruence between the phylogenetic relations among the leech hosts and symbionts, support the obligate and "ancient" nature of *Pr. siddallii* as a nutritional symbiont of *Haementeria* leeches. Nonetheless, *Pr. siddallii* has only ever been explored in a clade of South American and Mexican leech species, leaving the identity of the symbionts of its sister clade of Central and South American *Haementeria* species unknown.

In this work, we describe the serendipitous discovery of a novel nutritional symbiont belonging to the *Pluralibacter* genus in *Haementeria* leeches. Through whole-genome sequencing and genome-based metabolic inference, we show that this novel symbiont displays clear signs of an obligate nutritional-based association accompanied by continuous vertical transmission. We corroborate that both *Pr. siddallii* and the novel *Pluralibacter* symbiont show convergent retention of biosynthetic pathways for B vitamins, supporting the equivalent roles they play to complement their hosts' diet.

## MATERIALS AND METHODS

### Leech collection, DNA extraction, and sequencing

*Haementeria* sp. individuals were collected in Ciénega de las Macanas, Provincia de Herrera, Panamá (6 June 2017; 8.10303 N, 80.593974 W) by Sebastian Kvist, Danielle de Carle, and Alejandro Oceguera-Figueroa under permit number SE/A-61-17 (Ministerio de Ambiente de Panamá). Voucher specimens were deposited in the *Colección Zoológica Dr. Eustorgio Méndez* (CoZEM; https://www.gorgas.gob.pa/coleccion-cientifica/) from the *Instituto Conmemorativo Gorgas de Estudios de la Salud* (Panamá) under voucher numbers COZEM-ANN-HIR-001 and COZEM-ANN-HIR-002. Upon expert inspection of these individuals, it was concluded that they putatively represent specimens of a new leech species that remains to be described. The bacteriomes of two individuals were dissected, and total DNA was extracted using a commercial extraction kit (DNeasy Blood & Tissue Kit, Qiagen, Hilden, Germany), following the manufacturer's instructions for the purification of total DNA from animal tissues. DNA libraries were constructed using the NGS Nextera FLEX DNA library preparation kit (Illumina Inc., San Diego, CA) according to the manufacturer's protocol, and total DNA was multiplexed together with 11 other samples and sequenced on a single lane on the HiSeqX platform [150 base pairs (bp) paired end].

### Genome assembly and annotation

Sequencing reads were right-tailed-clipped (quality threshold of 20) using FASTX-Toolkit v0.0.14 (http://hannonlab.cshl.edu/fastx_toolkit/, last accessed 11 December 2023). We then removed reads containing undefined nucleotides ("N") and those shorter than 75 bp using PRINSEQ v0.20.4 (29). The resulting filtered reads were then assembled using SPAdes v3.10.1 (30) (-kmer 55,77,99 -only-assembler), and contigs with a coverage lower than 100 were discarded. The remaining contigs were binned using results from a BlastX v2.11.0 (31) search (best hit per contig) against a database consisting of the proteome of *Helobdella robusta* as well as that of a selection of bacterial strains (Table S1). *H. robusta* was selected for binning of leech-derived contigs given that, from the currently available leech genomes, it is the one originating from the closest relative to *Haementeria* spp. This binning resulted in 10 contigs of putative bacterial origin (coverage ≥1,000) and 13 assigned to the mitochondrial bin (coverage ≥100). The completeness of molecules assigned to both bacteria and mitochondrion was corroborated by both manual inspection of contigs showing a higher coverage than the background (i.e., host contigs), using the BlastX webserver (vs nr) and inspection of assembly graphs. In no case did we find any additional contigs of putative bacterial origin. The aforementioned binned contigs were used for mapping back the reads from *Haementeria* sp. to the two bins (endosymbiont and mitochondrion) using Bowtie2 v2.5.0 (32). This was followed by re-assembly of the mapped reads using SPAdes as described above. This resulted in a single mitochondrial contig and seven contigs (with overlapping ends) belonging to the putative bacterial endosymbiont.

Draft annotation of the genomes was done using MITOS v6b33f95 (33) and Prokka v1.14.16 (34). This draft annotation was followed by the manual curation of the protein-coding genes' coordinates as well as an update to the product naming using UniProtKB (35), Pfam v34 (36), and InterProScan (37). Annotations for non-coding RNAs were refined using Infernal v1.1.4 (38), with the Rfam database v14.1 (39), tRNAScan-SE v2.0.9 (40), and ARAGORN v1.2.41 (41). Pseudogenes were manually identified through online BlastX searches of the intergenic regions as well as through BlastP, DELTA-BLAST (42), and domain searches of the predicted open reading frames. Proteins were considered to be putatively functional if all essential domains for the function were found, if a literature search supported the truncated version of the protein as functional in a related organism, or if the predicted protein displayed truncations but retained identifiable domains. Details of the results of these searches are captured in the annotation files. All manual curation was done using UGENE v1.34.0 (43).

The fully annotated contigs underwent metabolic annotation in Pathway Tools v22.0 (44) based on the BioCyc and MetaCyc databases (45). Contigs were also submitted to the KEGG Automatic Annotation Service (46). After automatic reconstruction, manual curation of the databases was done by comparing to known reactions and complexes present in both BioCyc and KEGG. Global metabolic reconstruction was manually edited with Procreate v5.3.6.

## Phylogenetic analysis

For phylogenetic placement of the newly sequenced *Haementeria*-associated endosymbiont and informed by the 16S identity of it vs available sequences in NCBI's nr, we downloaded 147 representative genomes from *Enterobacteriaceae* and *Erwiniaceae*. Next, OrthoFinder v2.5.4 (47) was used to group predicted protein-coding genes across selected genomes in families of orthologous proteins. We then subset those consisting exclusively of single-copy core genes (i.e., present across all genomes), which were then individually aligned using MAFFT v7.490 (-maxiterate 1000 -localpair; Katoh and Standley (48)] and fed to Gblocks v0.91b (49) for the removal of divergent and ambiguously aligned blocks. The resulting alignment was then concatenated, and phylogenetic inference was done using IQtree v1.6.12 [LG4X+I; Minh et al. (50)] with 1,000 UltraFast bootstrap replicates (51). A tree plot was visualized and exported for editing using Figtree v1.4.4 (https://github.com/rambaut/figtree, last accessed 11 December 2023).

For phylogenetic placement of *Haementeria* sp., we followed Oceguera-Figueroa (52) and collected *cox1* gene sequences from representatives of *Placobdella*, *Helobdella*, and *Haementeria* genera. Sequences were aligned using MAFFT as described above. Phylogenetic inference was performed using MrBayes v3.2.7 [GTR+I+G4; Ronquist et al. (53)] running two independent analyses with four chains each for up to 1,000,000 generations and checked for convergence (average standard deviation below 0.001) with a burn-in of 25%. Tree visualization and exporting were done as described above.

## COG assignment and functional profiles

In order to investigate the differences in functional profile between the symbiont of *Haementeria* sp. and its closest free-living relatives (*"Pluralibacter*[=*Enterobacter*] *lignolyticus"* strain SCF1 and *Pluralibacter gergoviae* strain FDAARGOS 186), clusters of orthologous groups (COG) categories were assigned to all predicted coding sequences (CDSs) with the WebMGA web server (54). To assess the functional divergence of the novel leech symbiont from their free-living relative strains, the mean per COG category frequency was calculated for the latter and subtracted from the given category for both the leech symbiont and *Pluralibacter* strains as in reference (27). The results were visualized by means of a heatmap plotted in R (55) using the function *pheatmap*.

All figures were edited and exported using Inkscape v1.1.2 (https://inkscape.org, last accessed 11 December 2023), unless otherwise stated.

## RESULTS AND DISCUSSION

### Genome and phylogenetic placement of a novel *Haementeria* symbiont

As a result of whole-genome sequencing of the novel *Haementeria* sp. COZEM-ANN-HIR-001/002 (hereafter *Haementeria* sp.), a full mitochondrial genome was assembled. Using previously published mitochondrial *cox1* gene sequences, we were able to confidently place the yet-undescribed *Haementeria* sp. within the clade of Central and South American species, as sister to the clade made up of *Haementeria paraguayensis* and *Haementeria lutzi* (Fig. 1). The phylogenetic placement of the novel *Haementeria* species, thus, provided an opportunity to explore the identity and genomics of the symbiotic partner present in (at least one member of) the clade of *Haementeria* leeches sister to that of *Providencia*-hosting species.

The genome assembly of the novel bacterial symbiont sequenced from *Haementeria* sp. resulted in seven partially overlapping contigs (i.e., joined by end sequences but not

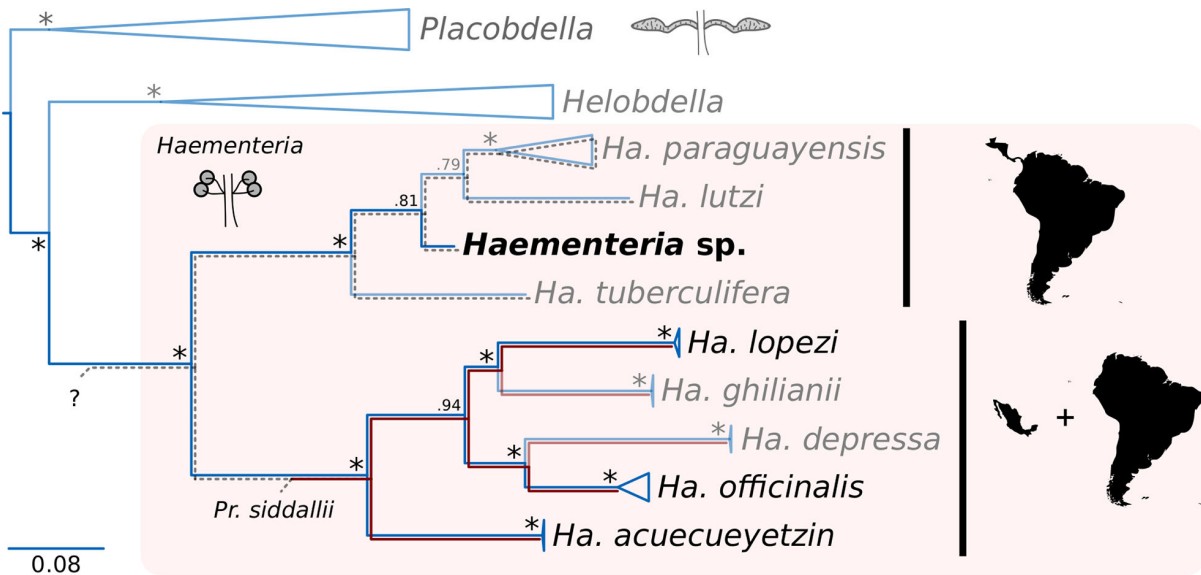

**FIG 1** Phylogenetic tree displaying relationships among *Haementeria* species, using *Placobdella* spp. and *Helobedella* spp. as outgrous. Diagrams of the bacteriomes of *Haementeria* and *Placobdella* are shown. Red lines indicate the known presence of the *Pr. siddallii* and its co-divergence with *Haementeria* hosts. Gray dotted lines indicate the presence of a bacteriome but unknown identity of the nutritional symbiont ("?"). In black font, members of *Haementeria* with sequenced endosymbiont genomes. In bold, the novel *Haementeria* sp. is highlighted. Numbers at nodes represent posterior probabilities of the bipartitions ("*" = 1).

resolved) adding up 1.18 mega base pairs (Mbp) with an average coverage of 1,030×. A BLASTN web search using the 16S sequence of the newly identified symbiont vs the NCBI's non-redundant nucleotide collection suggested a close relationship with the *Pluralibacter* genus (*Gammaproteobacteria: Enterobacteriaceae*). The novel symbiont encodes for a total of 846 intact proteins and retains 78 putative pseudogenes, whose products are either interrupted by a premature stop codon or are eroded to a point where essential domains are missing. When compared to the *Pr. siddallii* symbionts of other *Haementeria* (Table 1), it becomes evident that both lineages have experienced genome reduction similar to that of other strictly vertically transmitted symbionts of arthropods (8, 9, 56–58). However, the novel symbiont of *Haementeria* sp. displays a more moderate degree of genome reduction when compared to *Pr. siddallii* symbionts, namely, taking into account the G + C content, number of CDSs, RNA genes, and pseudogenes. This observed difference suggests that the symbiotic relation of the novel symbiont and *Haementeria* is younger than that of *Pr. siddallii* and their hosts. This would follow the prediction from comparative genomics that older more established symbionts

**TABLE 1** General genomic characteristics of *Haementeria* endosymbionts[a]

| | Pl. lignolyticus | | Pr. rettgeri | Pr. siddallii | | |
| --- | --- | --- | --- | --- | --- | --- |
| Bacterial strain | SCF1 | Novel symbiont | AR_0082 | PSAC | PSOF | PSLP |
| Genome size | 4.81 Mbp | ca. 1.19 Mbp | 844 kbp | 756 kbp | 844 kbp | 741 kbp |
| G + C content | 57.0% | 43.5% | 23.9% | 22.0% | 23.9% | 23.3% |
| CDSs | 4,399 | 846 | 628 | 610 | 628 | 612 |
| rRNAs | 22 | 9[b] | 3 | 3 | 3 | 3 |
| tRNAs | 82 | 42 | 33 | 33 | 33 | 34 |
| ncRNAs | NA | 10 | 2 | 2 | 2 | 2 |
| Pseudogenes | 50 | 71 | 5 | 4 | 5 | 5 |
| Mobile elements | Yes | None | Yes | None | None | None |

[a]PSAC, PSOF, and PSLP denote the *Pr. siddallii* strains associated with *Haementeria acuecueyetzin*, *Ha. officinalis*, and *Haementeria lopezi*, respectively. NA, not available in annotation.
[b]Calculated from the coverage of the rRNA operon contig.

have a highly degenerated but stable genome (as *Pr. siddallii*), while more recent ones tend to display larger and less degenerated ones [as the novel symbiont does (8, 9)].

In order to infer the phylogenetic origin of this novel *Haementeria*-associated symbiont and in light of the possible close relationship to the *Pluralibacter* genus, we performed phylogenetic reconstruction using the predicted proteomes of representatives from the *Enterobacteriaceae* and *Erwiniaceae* (Fig. 2; Fig. S1). The phylogenetic reconstruction clearly recovered the novel symbiont nested within the *Pluralibacter* genus together with *Pl. lignolyticus* and *Pl. gergoviae*. Similarly, the 16S rRNA gene identity and average nucleotide identity values, when compared to the other sequenced representatives of the genus (between 94.03–95.13 and 74.19–79.48, respectively), support this symbiont representing a novel species within the *Pluralibacter* genus (59–61). The phylogenetic reconstruction also evidenced a long branch leading to the novel *Pluralibacter* symbiont, a feature of many long-term vertically transmitted endosymbiont lineages, which reflects an accelerated evolution as a result of clonality and genetic drift (3, 4, 56, 57, 62).

## Metabolic capabilities of the novel *Pluralibacter* symbiont of *Haementeria* sp.

In order to gain insight into the genomic reduction and metabolic capabilities of the novel *Pluralibacter* symbiont of *Haementeria* sp., we performed clustering of their protein-coding genes and a COG category analysis to infer the functional profile divergence of the novel symbiont with respect to free-living *Pluralibacter* spp. (Fig. S2). Contrasting *Pr. siddallii*, whose genetic repertoire is almost exclusively a subset of that of its free-living relatives (27), the novel *Pluralibacter* symbiont retains a rather large number of strain-specific genes (129). This fact suggests that this symbiont likely evolved from within a yet-unknown *Pluralibacter* species, consistent with only two species known within the genus. The two free-living *Pluralibacter* spp. retain a highly similar COG functional profile with some differences in categories *G*, *T*, *W*, and *X*; translating to differences in their carbohydrate transport and metabolism, signal transduction mechanisms, extracellular structures, and mobile element content. On the other hand, the novel *Pluralibacter* symbiont shows a distinct profile, with functions such as those involved in cellular maintenance (*J* and *L*), cellular processes and signaling (*D*, *M*, and *O*), and metabolism (*C*, *F*, and *H*) overrepresented when compared to the free-living *Pluralibacter* spp. This latter category includes genes involved in co-enzyme transport and metabolism (*M*), which encompasses all genes involved in B vitamin biosynthesis and is highly retained in the novel *Pluralibacter* symbiont. This pattern is convergent to that observed in *Pr. siddallii* and other nutritional symbionts from obligate blood feeders (27, 57, 58, 63, 64), evidencing a convergent genome reduction among these symbiotic lineages.

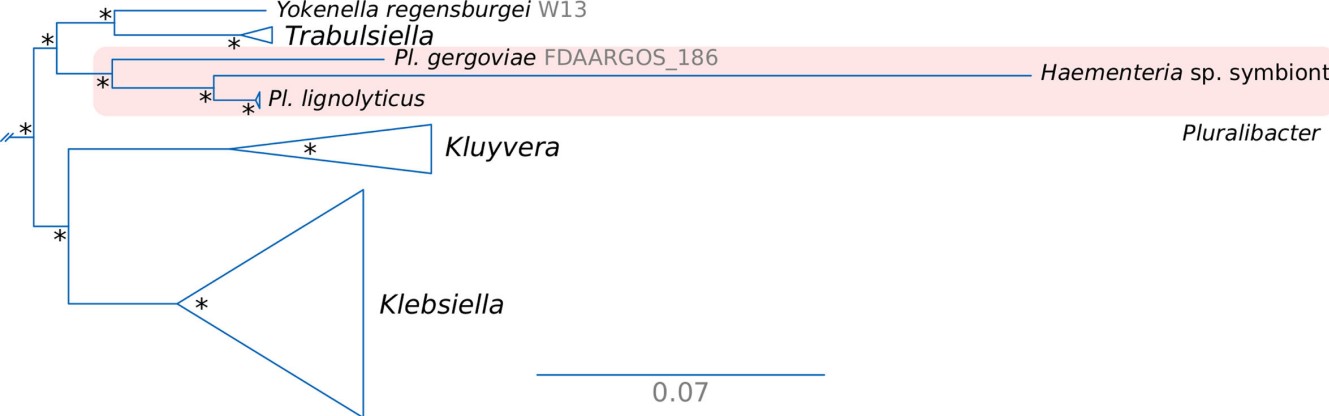

**FIG 2** Phylogenetic placement of the novel symbiont of *Haementeria* sp. Excerpt of maximum-likelihood phylogenetic reconstruction of *Enterobacteriaceae* representatives. The *Pluralibacter* clade is shaded in red. An asterisk at nodes represents an UltraFast bootstrap support of 100%.

The metabolic reconstruction of *Pl. haementeriicola* (Fig. S3) indicates that this bacteria can achieve facultative anaerobic respiration, coding for two types of NADH-dehydrogenases [NADH:quinone oxidoreductase I and the Na(+)-translocating NADH-quinone reductase NQR, which could pump H⁺ and Na⁺ protons, respectively], a fumarate reductase, and a cytochrome bd-I ubiquinol:oxygen oxidoreductase. This is in stark contrast to *Pr. siddallii*, which only preserves genes to perform anaerobic respiration (27). All the main enzymes for the central carbohydrate metabolism are present; thus, the endosymbiont would achieve the metabolism of glucose until the production of pyruvate that is then oxidized to obtain acetyl-coA, which then enters the tricarboxylic acid cycle. Similarly to *Pr. siddallii*, it preserves intact pathways to synthesize all purines and pyrimidines *de novo*.

Regarding the retention of B vitamin biosynthetic genes, which would theoretically compensate for its host's deficient blood-based diet, the novel *Pluralibacter* symbiont preserves almost-intact pathways for all B vitamin biosynthetic genes (Fig. 3). The one notable exception is the absence of *nudB*. However, the *nudB* gene is not universally conserved across endosymbionts of blood-feeding organisms (27, 28, 57). In fact, it has been shown that many phosphatases display wide-range substrate specificities in *Escherichia coli* (65, 66), suggesting that other phosphatase(s) encoded in the symbiont's genome might be fulfilling a similar role to *nudB*. In contrast to *Pr. siddallii*, the novel *Pluralibacter* symbiont could synthesize NAD from the import of nicotinate and could still synthesize pantothenate (vitamin B5) from aspartate and valine. In addition, the novel symbiont retains redundant enzymes for some metabolic steps of the B vitamin biosynthetic pathways, unlike *Pr. siddallii* and other well-established symbionts (28, 67–69). These metabolic features all support a younger age for the association of *Haementeria* leeches with *Pluralibacter* compared to that with *Providencia*.

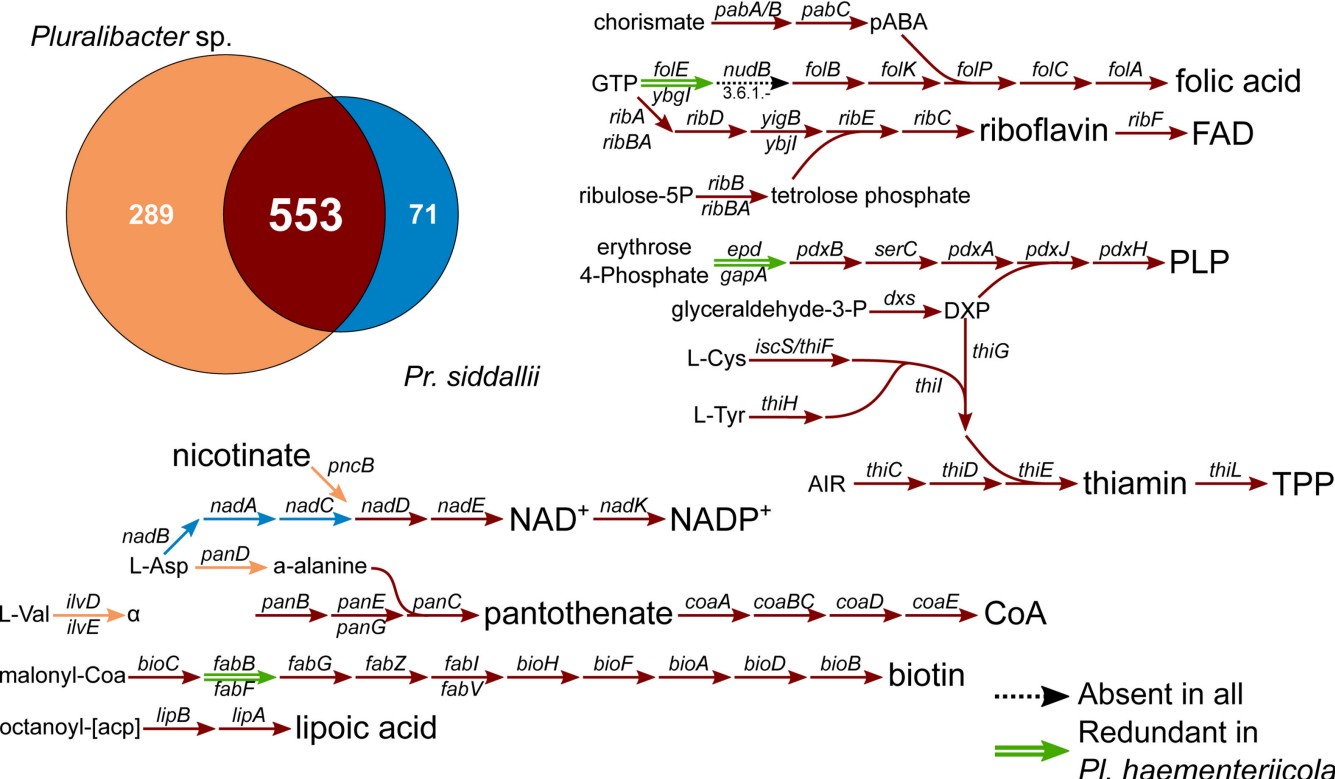

**FIG 3** Shared genes and B vitamin pathways of *Haementeria*-associated nutritional symbionts. At the top left, a Venn-like diagram displaying the results of *OrthoMCL* clustering of the predicted proteomes of *Pr. siddallii* from *Ha. officinalis* and the novel *Pluralibacter* symbiont. Diagram of the B vitamin biosynthetic pathways. Arrows connect metabolites, and names on them indicate the gene coding for the enzyme involved in the enzymatic step.

Regarding amino acid biosynthesis, it has lost the ability to *de novo* synthesize most of the essential amino acids, except for arginine and phenylalanine (from chorismate), as well as non-essential ones (Fig. S3). However, it retains importers for branched-chain amino acids (isoleucine, leucine, and valine; BrnQ), aspartate (DcuA, DauA), serine and threonine (SstT), and aromatic amino acids (phenylalanine, tryptophan, and tyrosine; AroP). We found evidence that the *Pluralibacter* symbiont was formerly able to code for a methionine-specific ABC transporter complex (MetNIQ), from which a pseudogenized *metN* is retained. Enzymes involved in tryptophan, histidine, and methionine biosynthesis are completely absent. Therefore, the symbiont would largely depend on external sources for the supply of most amino acids. Since after water, proteins are the most abundant organic compound in blood (70), the host's diet could represent such an external source to solve both host and endosymbiont essential amino acid requirements.

## '*Candidatus* Pluralibacter haementeriicola' *sp. nov.*

'*Candidatus* Pluralibacter haementeriicola' (hae.men.te.ri.i'co.la. N.L. fem. n. *Haementeria*, a leech genus; L. masc./fem. n. suff. -cola, inhabitant, dweller; N.L. masc. n. *haementeriicola*, inhabiting *Haementeria* leeches).

We propose the specific name '*Candidatus* Pluralibacter haementeriicola' for the lineage of enterobacterial endosymbionts from the *Pluralibacter* genus (*Enterobacterales: Enterobacteriaceae*) hitherto exclusively found as a bacteriocyte-associated nutritional endosymbiont of *Haementeria* sp. COZEM-ANN-HIR-001/002. Using available genomic data, the closest relative of this symbiont lineage is "*Pluralibacter*(=[*Enterobacter*]) *lignolyticus*." The only available genome from '*Candidatus* Pluralibacter haementeriicola' displays clear evidence of strong genome reduction and, based on a genome-based metabolic reconstruction, functions as a nutritional endosymbiont providing B vitamins for its leech host. Given the current lack of microscopic imaging of the bacteriomes of its host, the cellular shape and specific localization within these organs are hitherto unknown.

## Conclusion

Our results provide strong evidence for '*Ca.* Pluralibacter haementeriicola' (hereafter *Pl. haementeriicola*) fulfilling the equivalent role to *Pr. siddallii* as a B vitamin provider for its *Haementeria* leech host. Moreover, the phylogenetic positioning of *Pl. haementeriicola*, its genomic characteristics, and metabolic capabilities suggest an established but younger symbiotic association than that of *Pr. siddallii* with *Haementeria* leeches. While obligate symbiont replacement has been well studied in arthropods (12, 15, 71), other systems have not undergone such a deep study. The aforementioned features from *Pl. haementeriicola*, as well as its host phylogenetic position, suggest that *Pl. haementeriicola* replaced a *Pr. siddallii* symbiont at least in its host species. We expect that further information from endosymbionts of leeches from the same Central and South American clade of *Haementeria* will clarify whether this novel symbiosis represents a recent symbiont substitution or a more ancient association. Furthermore, the future availability of high-quality leech genomes will undoubtedly provide useful insights into the host factors driving these host–microbe symbioses and their turnover.

## ACKNOWLEDGMENTS

This paper is part of the requirements for obtaining a Doctoral degree at the Posgrado en Ciencias Biológicas, UNAM, of V.M.S.-J. We thank the Posgrado en Ciencias Biologicas, UNAM, and CONACyT for providing a scholarship to V.M.S.-J. A.M.-M. has received funding from European Union's Horizon 2020 Research and Innovation Programme under the Marie Sklodowska-Curie Grant Agreement No. [840270] (LEECHSYMBIO). The authors thank Prof. Dr. Thomas Rattei and his team for maintaining the Life Science Compute Cluster (LiSC; https://lisc.univie.ac.at) that was used for computational analyses.

Microbiology Spectrum

The funders had no role in the study design, data collection and analysis, decision to publish, or preparation of the manuscript.

This project was funded by grants from PAPIIT-UNAM-IN213520 to A.O.-F., NSERC to S.K., and a Peer Review Grant from the Royal Ontario Museum to S.K.

## AUTHOR AFFILIATIONS

[1]Posgrado en Ciencias Biológicas, Universidad Nacional Autónoma de México, Ciudad de México, Mexico

[2]Departamento de Zoología, Instituto de Biología, Universidad Nacional Autonoma de México, Ciudad de México, Mexico

[3]Department of Natural History, Royal Ontario Museum, Toronto, Ontario, Canada

[4]Department of Ecology and Evolutionary Biology, University of Toronto, Toronto, Ontario, Canada

[5]Centre for Microbiology and Environmental Systems Science, University of Vienna, Vienna, Austria

## PRESENT ADDRESS

Sebastian Kvist, Swedish Museum of Natural History, Stockholm, Sweden

## AUTHOR ORCIDs

Víctor Manuel Sosa-Jiménez http://orcid.org/0009-0009-4201-3399
Alejandro Manzano-Marín http://orcid.org/0000-0002-0707-9052

## FUNDING

| Funder | Grant(s) | Author(s) |
|---|---|---|
| Universidad Nacional Autónoma de México (UNAM) | PAPIIT-UNAM-IN213520 | Alejandro Oceguera-Figueroa |
| European Commission (EC) | 84027 | Alejandro Manzano-Marín |

## DATA AVAILABILITY

All auxiliary files and data for other analyses as well as the annotated genome of *Pl. haementeriicola* and its host mitochondrion are available online at https://doi.org/10.5281/zenodo.10091738. Newly sequenced and annotated genomes are also deposited at the European Nucleotide Archive (ENA) under project number PRJEB74922.

## ADDITIONAL FILES

The following material is available online.

### Supplemental Material

**Supplemental material (Spectrum04286-23-s0001.pdf).** Fig. S1 to S3.
**Table S1 (Spectrum04286-23-s0002.xlsx).** Organisms used for taxonomic binning of reads and comparative analyses.

### Open Peer Review

**PEER REVIEW HISTORY (review-history.pdf).** An accounting of the reviewer comments and feedback.

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
