## [Reviewer comments · Microbiology Spectrum]

Microbiology Spectrum

Discovery of a novel symbiotic lineage associated with a haematophagous leech from the genus *Haementeria*

Víctor Sosa-Jiménez, Sebastian Kvist, Alejandro Manzano-Marín, and Alejandro Oceguera-Figueroa

Corresponding Author(s): Alejandro Manzano-Marín, Universitat Wien

Review Timeline:

Submission Date:	December 26, 2023
Editorial Decision:	February 28, 2024
Revision Received:	April 26, 2024
Accepted:	April 29, 2024

Editor: Rosario Gil

Reviewer(s): The reviewers have opted to remain anonymous.

Transaction Report:

DOI: <https://doi.org/10.1128/spectrum.04286-23>

Re: Spectrum04286-23 (A new guest to the blood feast: a novel symbiotic lineage associated with a haematophagous leech from the genus *Haementeria*)

Dear Dr. Alejandro Manzano-Marín:

Thank you for the privilege of reviewing your work. Your manuscript has been reviewed by two experts on the subject, both of which agree that it needs some improvement prior being acceptable for publication in Microbiology Spectrum.

First of all, I detected that your work has been submitted as an "Observation" but it does not fulfill the conditions to be so considered, based on the Instructions to authors: "Observations are short descriptions of research results of interest to the broad microbiology community, e.g., reports of a new type of organism, a new organelle, a new association of microbes and disease, etc. The body of an Observation may have paragraph lead-ins. Authors should include an abstract of 250 words or fewer as well as an Importance section of 150 words or fewer, providing a nontechnical explanation of why the work was undertaken. 1,200 words with a maximum of 2 figures and 25 references." Consequently, I recommend you to change the classification to be considered as a "Research article". This will allow you to provide more information in the Introduction section and to add the additional analyses suggested by the reviewers.

Below you will find instructions from the Spectrum editorial office, and the reviewer comments.

Revision Guidelines

Sincerely,
Rosario Gil

Reviewer #2 (Comments for the Author):

In their article, Sosa-Jiménez et al. highlighted the existence of a new species of nutritional symbiont associated with an unknown leech species of the genus *Haementeria*. This is a short "Observation" type article that is overall very well written and sets the scene for further future study on the associations that leeches can evolve with symbiotic nutritional bacteria. In my opinion, this is a very exciting subject that is still in its infancy. However, the subject of metabolic complementarity between host and symbiont is now a little hackneyed, especially through in silico approaches (so much has already been done with insects). That being said, the scientific work has been well done. I personally look forward to seeing this topic develop further in the future, particularly the transmission and developmental biology aspects, with further research into the genetic basis that two different phyla (Annelida versus Arthropoda) may have in common for the development of bacteriocytes and the control of symbionts. Very, very exciting: this is the holy grail the authors should aim for quickly. That said, I have a few comments to improve the article.

Title

The expression "A new guest to the blood feast" suggests that insect and bacterium are on the same level, or that a new bacterium comes to sit at another bacterium's table to share something (as in di-symbiotic systems). This seems to be false: there is no sharing of meals, but a bacterium that helps its host leech to access the nutrients it lacks in its blood meal. "Discovery of a novel symbiotic lineage associated with a haematophagous leech from the genus *Haementeria*" is enough to describe the study as it is (no need to use racy titles that don't exactly match reality).

Abstract

L20-24: The sentence "Nonetheless, given a lack of molecular investigations, the identity of the symbiont housed in the bacteriomes of its sister clade of Central and South American congeners remained unknown" could be better formulated.

I think the authors should phrase it this way: "Candidatus *Providencia siddallii* has been identified as the obligate nutritional endosymbiont of a monophyletic clade of Mexican and South American *Haementeria* spp. However, the *Haementeria* genus includes a sister clade of congeners from Central and South America, and it is not known whether *P. siddallii* symbionts are also pervasive in this clade or whether its members have evolved partnerships with other bacterial species."

L44-49: I disagree. The fact that *Pluralibacter* seems to have formed a more recent relationship with *Haementeria* does not necessarily mean that it has replaced *Providencia*. It doesn't seem right to say that the association between *Haementeria* and *Providencia* is prone to breakdown. The authors may suggest the replacement of *Providencia* by *Pluralibacter*, but they cannot conclude that the *Haementeria*-*Providencia* association is prone to breakdown: It is a hypothesis, but it can't be a factual conclusion, especially as the study concerns only one case of association in the clade studied (it is very limited). In comparison, the conclusion (p. 8) is more honest and balanced.

L47: *Hamentaria* is misspelled.

Introduction

The introduction is short, to the point and well written. The three evolutionary scenarios are well summarized. However, I sometimes find references lacking. References are missing, for example in the second part of the first paragraph (evolutionary scenarios) and the first part of the second paragraph (references for sap-feeding insects).

Also: make sure you use the most relevant references. E. g. L54: I don't think that Gu et al. 2023 is an accurate reference here because it is about an invasive heritable symbiont but not especially about the nutritional side of endosymbiosis (you should use references directly linked to primary (obligate) symbiosis such as *Buchnera*, *Carsonella*, *Portiera* and many others). Please consider the relevance of the references you use.

L77: the authors should avoid clumsy puns ((in)famous) in scientific articles (science is a serious business that can be fun, but scientific publication is not a comedy stage).

Materials and methods

It's a pity that the authors didn't invest in long-read sequencing to circularize the *Pluralibacter* genome (isn't it possible to close these 10 contigs?). After all, the work only concerns one sample, and it wouldn't have cost that much more. This would have offered good opportunities for comparative analyses of genome structure between mutualistic endosymbiotic strains and free-living strains (rearrangements, etc.). Of course, for functional analyses, as in this case, it's enough.

The genomic analyses were performed appropriately. The biosynthetic capacity of the *Pluralibacter* symbiont for vitamin B was compared with that of *Providencia*. This is logical given the data recovered. However, the limiting factor in this type of analysis is the absence of genomic data from the host to determine exactly what the symbiont and host bring into the system, and to deduce what does and does not come from the "input feed (blood)". This is a recurring limitation in this type of analysis, whereas to be complete and more assertive in the conclusions, the genomes of the different members of the system are desirable. This is not a limit to the publication of the article, of course, but should be seen as a comment to push the authors to go one step further when using genomic inference. I therefore suggest that authors open a small one-line perspective to present the limitations and draw attention to the possibility of having the host genome in the future. This will encourage other research groups to move in this direction, rather than limiting themselves to extrapolations based on genomic inferences based solely on symbiont genomes (after all, host genomes are not fixed, they evolve and there is also inter- and intra-species variability, an aspect which, in my opinion, is too often elided).

Where are the genomic sequences stored? In which databases have they been deposited? It doesn't seem to me that this is indicated.

Results and discussion

L188-194: I don't understand. Is the host genome sequenced? Where is the reference if it is? Why haven't these data been used to refine the metabolic analyses and offer the community a complete view of the complementarity between host and symbiont? This would have been more judicious than relying solely on the new symbiont's ability to synthesize the vitamin. The authors' hypothesis is undoubtedly correct, but it would be necessary to relate this data to metabolic information from the host (a good host-symbiont genomic inference scheme, at least for B vitamins). It may be asking too much, but it's important to show what the leech still has in its belly in terms of biosynthesis capabilities (which genes are present and which are absent).

L224-224: What is the reason for this long branch characteristic (please complete with "due to..." to finish the sentence)?

L273: *Providencia* is misspelled. Please, reread the manuscript to check spelling everywhere.

The conclusion is well balanced.

Overall, the results are of good quality, even if they don't add anything new to the field of nutritional symbiosis. They do, however, draw attention to a very promising model that should be exploited with an Evo-Devo approach.

Reviewer #3 (Comments for the Author):

The manuscript by Sosa-Jiménez et al., entitled "A new guest to the blood feast: a novel 2 symbiotic lineage associated with a haematophagous leech from the genus *Haementeria*" is an interesting example of how endosymbiosis has allowed insects to adapt to nutrient-deficient diets with the help of bacteria from different clades, which undergo a convergent evolution to be able to provide the same essential products to related hosts feeding on similar diets (blood in this case) in different planet regions.

I find the topic very interesting, and I have no major concerns about the manuscript as a whole, but I would like to make some comments that might help to clarify or improve some parts of the text.

The Introduction contains an excess of references which are not always relevant or the most appropriate. It might be interesting to include some previous reviews instead, since all work on the subject cannot be cited.

L34: Considering that the hypothesis of endosymbiont replacement is the main conclusion in this manuscript, the presentation of this evolutionary outcome should be more extensively presented in the introduction for those readers that would be not so familiar with the topic.

L122: "Surviving" is a rare word choice to talk about retained contigs.

L123: *Helobdella robusta* appears only once here in the whole text. I guess the authors use this leech of the family Glossiphoniidae because its genome has been sequenced, but they should explain so for the readers. It might be interesting to talk about it in the introduction.

L134: The final number of contigs corresponding to the endosymbiont genome are not clear. The authors mention they are 10 contigs with overlapping ends (line 134) or 7 partially-overlapping contigs (line 196). Why the circular structure could not be resolved? Couldn't that be solved with long reads with PacBio or Nanopores, or even using directed PCR and Sanger sequencing?

L159: Please review sentence "We then those consisting..."

L215: It would be interesting to provide a phylogenetic reconstruction showing the distance between the two endosymbiotic bacteria found in the family Glossiphoniidae. Is there a single *Providencia siddallii* strain previously described or there are different strains in different related insects?

Table 1: The amount of rRNA genes is calculated based on the coverage of the rRNA operon contig. Does that mean that this is one of the reasons to be unable to close the genome?

L273: Correct "Providencia"

References: Please check. I detected that reference 9 and 77 are the same.

In their article, Sosa-Jiménez et al. highlighted the existence of a new species of nutritional symbiont associated with an unknown leech species of the genus *Haementeria*. This is a short “Observation” type article that is overall very well written and sets the scene for further future study on the associations that leeches can evolve with symbiotic nutritional bacteria. In my opinion, this is a very exciting subject that is still in its infancy. However, the subject of metabolic complementarity between host and symbiont is now a little hackneyed, especially through *in silico* approaches (so much has already been done with insects). That being said, the scientific work has been well done. I personally look forward to seeing this topic develop further in the future, particularly the transmission and developmental biology aspects, with further research into the genetic basis that two different phyla (Annelida versus Arthropoda) may have in common for the development of bacteriocytes and the control of symbionts. Very, very exciting: this is the holy grail the authors should aim for quickly. That said, I have a few comments to improve the article.

Title

The expression “A new guest to the blood feast” suggests that insect and bacterium are on the same level, or that a new bacterium comes to sit at another bacterium’s table to share something (as in di-symbiotic systems). This seems to be false: there is no sharing of meals, but a bacterium that helps its host leech to access the nutrients it lacks in its blood meal. “Discovery of a novel symbiotic lineage associated with a haematophagous leech from the genus *Haementeria*” is enough to describe the study as it is (no need to use racy titles that don’t exactly match reality).

Abstract

L20-24: The sentence “Nonetheless, given a lack of molecular investigations, the identity of the symbiont housed in the bacteriomes of its sister clade of Central and South American congeners remained unknown” could be better formulated.

I think the authors should phrase it this way: “Candidatus *Providencia siddallii* has been identified as the obligate nutritional endosymbiont of a monophyletic clade of Mexican and South American *Haementeria* spp. However, the *Haementeria* genus includes a sister clade of congeners from Central and South America, and it is not known whether *P. siddallii* symbionts are also pervasive in this clade or whether its members have evolved partnerships with other bacterial species.”

L44-49: I disagree. The fact that *Pluralibacter* seems to have formed a more recent relationship with *Haementeria* does not necessarily mean that it has replaced *Providencia*. It doesn’t seem right to say that the association between *Haementeria* and *Providencia* is prone to breakdown. The authors may suggest the replacement of *Providencia* by *Pluralibacter*, but they cannot conclude that the *Haementeria*-*Providencia* association is prone to breakdown: It is a hypothesis, but it can’t be a factual conclusion, especially as the study concerns only one case of association in the clade studied (it is very limited). In comparison, the conclusion (p. 8) is more honest and balanced.

L47: *Hamentaria* is misspelled.

Introduction

The introduction is short, to the point and well written. The three evolutionary scenarios are well summarized. However, I sometimes find references lacking. References are missing, for example in the second part of the first paragraph (evolutionary scenarios) and the first part of the second paragraph (references for sap-feeding insects).

Also: make sure you use the most relevant references. E. g. L54: I don't think that Gu et al. 2023 is an accurate reference here because it is about an invasive heritable symbiont but not especially about the nutritional side of endosymbiosis (you should use references directly linked to primary (obligate) symbiosis such as *Buchnera*, *Carsonella*, *Portiera* and many others). Please consider the relevance of the references you use.

L77: the authors should avoid clumsy puns ((in)famous) in scientific articles (science is a serious business that can be fun, but scientific publication is not a comedy stage).

Materials and methods

It's a pity that the authors didn't invest in long-read sequencing to circularize the *Pluralibacter* genome (isn't it possible to close these 10 contigs?). After all, the work only concerns one sample, and it wouldn't have cost that much more. This would have offered good opportunities for comparative analyses of genome structure between mutualistic endosymbiotic strains and free-living strains (rearrangements, etc.). Of course, for functional analyses, as in this case, it's enough.

The genomic analyses were performed appropriately. The biosynthetic capacity of the *Pluralibacter* symbiont for vitamin B was compared with that of *Providencia*. This is logical given the data recovered. However, the limiting factor in this type of analysis is the absence of genomic data from the host to determine exactly what the symbiont and host bring into the system, and to deduce what does and does not come from the "input feed (blood)". This is a recurring limitation in this type of analysis, whereas to be complete and more assertive in the conclusions, the genomes of the different members of the system are desirable. This is not a limit to the publication of the article, of course, but should be seen as a comment to push the authors to go one step further when using genomic inference. I therefore suggest that authors open a small one-line perspective to present the limitations and draw attention to the possibility of having the host genome in the future. This will encourage other research groups to move in this direction, rather than limiting themselves to extrapolations based on genomic inferences based solely on symbiont genomes (after all, host genomes are not fixed, they evolve and there is also inter- and intra-species variability, an aspect which, in my opinion, is too often elided).

Where are the genomic sequences stored? In which databases have they been deposited? It doesn't seem to me that this is indicated.

Results and discussion

L188-194: I don't understand. Is the host genome sequenced? Where is the reference if it is? Why haven't these data been used to refine the metabolic analyses and offer the community a complete view of the complementarity between host and symbiont? This would have been more judicious than relying solely on the new symbiont's ability to synthesize the vitamin. The authors' hypothesis is undoubtedly correct, but it would be necessary to relate this data to metabolic information from the host (a good host-symbiont genomic inference scheme, at least for B vitamins). It may be asking too much, but it's important to show what the leech still has in its belly in terms of biosynthesis capabilities (which genes are present and which are absent).

L224-224: What is the reason for this long branch characteristic (please complete with "due to..." to finish the sentence)?

L273: *Providencia* is misspelled. Please, reread the manuscript to check spelling everywhere.

The conclusion is well balanced.

Overall, the results are of good quality, even if they don't add anything new to the field of nutritional symbiosis. They do, however, draw attention to a very promising model that should be exploited with an Evo-Devo approach.

Associate Editor Recommendation:

Thank you for the privilege of reviewing your work. Your manuscript has been reviewed by two experts on the subject, both of which agree that it needs some improvement prior being acceptable for publication in Microbiology Spectrum.

First of all, I detected that your work has been submitted as an "Observation" but it does not fulfill the conditions to be so considered, based on the Instructions to authors: "Observations are short descriptions of research results of interest to the broad microbiology community, e.g., reports of a new type of organism, a new organelle, a new association of microbes and disease, etc. The body of an Observation may have paragraph lead-ins. Authors should include an abstract of 250 words or fewer as well as an Importance section of 150 words or fewer, providing a nontechnical explanation of why the work was undertaken. 1,200 words with a maximum of 2 figures and 25 references." Consequently, I recommend you to change the classification to be considered as a "Research article". This will allow you to provide more information in the Introduction section and to add the additional analyses suggested by the reviewers.

>> We appreciate the kind words. We had indeed decided upon submission as an "Observation" paper, as we viewed the work as a "short descriptions of research results of interest to the broad microbiology community" (reports of a new type of organism). After going through your recommendation and the reviewers' comments, we agree that it would present better as a "Research Article" rather than as an observation. We have thus made the necessary changes for the resubmission <<

Reviewers' Comments:

Reviewer 2:

In their article, Sosa-Jiménez et al. highlighted the existence of a new species of nutritional symbiont associated with an unknown leech species of the genus Haementeria. This is a short "Observation" type article that is overall very well written and sets the scene for further future study on the associations that leeches can evolve with symbiotic nutritional bacteria. In my opinion, this is a very exciting subject that is still in its infancy. However, the subject of metabolic complementarity between host and symbiont is now a little hackneyed, especially through in silico approaches (so much has already been done with insects). That being said, the scientific work has been well done. I personally look forward to seeing this topic develop further in the future, particularly the transmission and developmental biology aspects, with further research into the genetic basis that two different phyla (Annelida versus Arthropoda) may have in common for the development of bacteriocytes and the control of symbionts. Very, very exciting: this is the holy grail the authors should aim for quickly. That said, I have a few comments to improve the article.

>> We thank the reviewer for his/her kind words. We indeed agree the topic of obligate symbioses across leeches is still in its infancy, thus, we find ourselves still doing a lot of the groundwork of describing the nature and identity of the symbioses we are able to find, for example, those nutritional symbioses found in bacteriome-bearing leeches.

Through the genomic exploration we are gaining insight into the possible functionalities of these beyond the expected nutritional complementation they display, however, this is the topic of future work that is ongoing and yet incomplete. We agree with the reviewer that symbiont transmission and bacteriome development are very exciting topics, especially so when comparing these processes with those observed, and some well-described, examples from arthropods. If some ongoing project applications result successful, then we will be able to dive deeper into these processes. However, for now, this work is halted due to lack of funding. <<

Title

The expression "A new guest to the blood feast" suggests that insect and bacterium are on the same level, or that a new bacterium comes to sit at another bacterium's table to share something (as in di-symbiotic systems). This seems to be false: there is no sharing of meals, but a bacterium that helps its host leech to access the nutrients it lacks in its blood meal. "Discovery of a novel symbiotic lineage associated with a haematophagous leech from the genus *Haementeria*" is enough to describe the study as it is (no need to use racy titles that don't exactly match reality).

>> We appreciate the reviewer's comment. We do however not agree with this view and find the title's play of words appropriate to both call attention upon potential readers as well as to convey the main finding. We do however appreciate that not everyone is enthusiastic about these sort of titles. We have thus rephrased as "**Discovery of a novel symbiotic lineage associated with a haematophagous leech from the genus *Haementeria***" <<

Abstract

L20-24: The sentence "Nonetheless, given a lack of molecular investigations, the identity of the symbiont housed in the bacteriomes of its sister clade of Central and South American congeners remained unknown" could be better formulated.

I think the authors should phrase it this way: "Candidatus *Providencia siddallii* has been identified as the obligate nutritional endosymbiont of a monophyletic clade of Mexican and South American *Haementeria* spp. However, the *Haementeria* genus includes a sister clade of congeners from Central and South America, and it is not known whether *P. siddallii* symbionts are also pervasive in this clade or whether its members have evolved partnerships with other bacterial species."

>> We have rephrased as "However, the *Haementeria* genus includes a sister clade of congeners from Central and South America, where the presence or absence of the aforementioned symbiont taxon remains unknown." <<

L44-49: I disagree. The fact that *Pluralibacter* seems to have formed a more recent relationship with *Haementeria* does not necessarily mean that it has replaced *Providencia*. It doesn't seem right to say that the association between *Haementeria* and *Providencia* is prone to breakdown. The authors may suggest the replacement of *Providencia* by *Pluralibacter*, but they cannot conclude that the *Haementeria-Providencia* association is

prone to breakdown: It is a hypothesis, but it can't be a factual conclusion, especially as the study concerns only one case of association in the clade studied (it is very limited). In comparison, the conclusion (p. 8) is more honest and balanced.

>> We agree with the reviewer's view, and as he rightfully points out, a more nuanced statement is present in the conclusion. The statement in the abstract was not meant as a dishonest one, but rather one that clearly ended up formulated in an unintended way (one can get lost in revisions and versions). We think that in order to be able to say that "the *Haementeria-Pluralibacter* association is prone to breakdown", we would likely need to observe several instances of symbiont replacement and/or complementation (similar to what is observed in aphids and adelgids). We do know that *Haementeria*-symbiont can break down, as the bacteriome precedes both *Providencia* and *Pluralibacter*: thus, one of the two, or even an extinct symbiont lineage, was already present at the origin of this leech genus. We have thus reformulated the last paragraph of the abstract as follows: "We conclude that the once-thought stable associations between blood-feeding Glossiphoniidae and their symbionts (*i.e.* one bacteriome structure, one symbiont lineage) can breakdown, mirroring symbiont turnover observed in various arthropod lineages." <<

L47: Hamentaria is misspelled.

>> Corrected. <<

Introduction

The introduction is short, to the point and well written. The three evolutionary scenarios are well summarized. However, I sometimes find references lacking. References are missing, for example in the second part of the first paragraph (evolutionary scenarios) and the first part of the second paragraph (references for sap-feeding insects).

>> We appreciate the comment. However, references are not actually missing, but maybe it bears repeating or changing the placement of some of them. For example, reference 9 and 10 go into the evolutionary scenarios, but to avoid repetition, they were not repeated after the mention two sentences away. In regards to the references for sap-feeding insects, it is true that we did not include further ones. We have now adapted the placement of references and included two to refer to the nutritionally unbalanced content of essential amino acids and B vitamins in plant sap. We have also gone through the text and adapted or added citations. <<

Also: make sure you use the most relevant references. E. g. L54: I don't think that Gu et al. 2023 is an accurate reference here because it is about an invasive heritable symbiont but not especially about the nutritional side of endosymbiosis (you should use references directly linked to primary (obligate) symbiosis such as Buchnera, Carsonella, Portiera and many others). Please consider the relevance of the references you use.

>> It is absolutely true that Gu *et. al.* is an inaccurate citation here. We apologise for this incorrect inclusion. It was likely a remnant of an earlier version of the manuscript (when referring to the vertical transmission of a symbiont). It has now been removed. <<

L77: the authors should avoid clumsy puns ((in)famous) in scientific articles (science is a serious business that can be fun, but scientific publication is not a comedy stage).

>> Please let us assure you "(in)famous" is not a "clumsy pun". The use of parenthesis in written English is a legitimate way of collapsing two words that share either a suffix or prefix. Additionally, the use of both "famous" and "infamous" is intentional. On one side, leeches are famous (adj. having a widespread reputation, usually of a favourable nature) for a particular trait shared by most leeches (their blood-feeding habit). On the flip side, they are infamous (adj. having an extremely bad reputation) for this very same trait in medicine and in popular culture. Therefore, no, the use of "(in)famous" is in no way a "clumsy pun" but a statement about the nature of their fame. To make this more explicit we have separated the two words. <<

Materials and methods

It's a pity that the authors didn't invest in long-read sequencing to circularize the *Pluralibacter* genome (isn't it possible to close these 10 contigs?). After all, the work only concerns one sample, and it wouldn't have cost that much more. This would have offered good opportunities for comparative analyses of genome structure between mutualistic endosymbiotic strains and free-living strains (rearrangements, etc.). Of course, for functional analyses, as in this case, it's enough.

>> It is indeed a pity indeed not to count with a closed genome. When working with complex samples (novel species with very few individuals sampled on a single host), sampled in complex environments (remote places), one can only work with certain storage conditions (generally not suitable for long-read sequencing). Furthermore, as we believe the reviewer has noticed, most of the work (at least the costly part) has been conducted in Mexico, a country where funds are, to say the least, extremely limited. Thus, no funds were available to pay for the generation and sequencing of a long-read library. <<

The genomic analyses were performed appropriately. The biosynthetic capacity of the *Pluralibacter* symbiont for vitamin B was compared with that of *Providencia*. This is logical given the data recovered. However, the limiting factor in this type of analysis is the absence of genomic data from the host to determine exactly what the symbiont and host bring into the system, and to deduce what does and does not come from the "input feed (blood)". This is a recurring limitation in this type of analysis, whereas to be complete and more assertive in the conclusions, the genomes of the different members of the system are desirable. This is not a limit to the publication of the article, of course, but should be seen as a comment to push the authors to go one step further when using genomic inference. I therefore suggest that authors open a small one-line perspective to present the limitations and draw attention to the possibility of having the host genome in the future. This will encourage other research groups to move in this direction, rather than limiting themselves to extrapolations based on genomic inferences based solely on symbiont genomes (after all, host genomes are not fixed, they evolve and there is also inter- and intra-species variability, an aspect which, in my opinion, is too often elided).

>> Regarding this point, again, we would love to count for paired genomes of many *Haementeria*-symbiont pairs. However, this is not yet a reality and especially not for this work. The reviewer might be unaware that we are often working with immensely limited funds, meaning that novel sampling and any sort of sequencing is already a huge expenditure effort. Some of the authors from this work are involved in generating several high-quality leech genomes, but this is a matter of other unrelated projects. We have added a clarification of this in the conclusion "Furthermore, the future availability of high-quality leech genomes will undoubtedly provide useful insights into the host factors driving these host-microbe symbioses and their turnover." <<

Where are the genomic sequences stored? In which databases have they been deposited? It doesn't seem to me that this is indicated.

>> As indicated in the "Supplemental Material & Data Availability" section, the sequences are deposited at ENA, but were not yet accessioned. They are nonetheless available in the mentioned Zenodo repository. We have actually had huge problems with endosymbiont genomes and annotation of genes with likely transcriptional slippage. After more than a year of to and fro with the ENA helpdesk (where they repeatedly assured us the "10 nt intron" problem was solved) we decided to deposit as-is and attempt to correct it after deposition (not ideal, but see no other way of doing this). Given that both the host and symbiont taxa are new, a new taxon is to be requested from ENA before submission of genomic sequences. These both have been requested, but no response assigning a taxon has been received. Sequences will be uploaded and accessioned once the taxon received an ID. The project number under which the sequences will be available is PRJEB74922 (as stated in the updated aforementioned section). <<

Results and discussion

L188-194: I don't understand. Is the host genome sequenced? Where is the reference if it is? Why haven't these data been used to refine the metabolic analyses and offer the community a complete view of the complementarity between host and symbiont? This would have been more judicious than relying solely on the new symbiont's ability to synthesize the vitamin. The authors' hypothesis is undoubtedly correct, but it would be necessary to relate this data to metabolic information from the host (a good host-symbiont genomic inference scheme, at least for B vitamins). It may be asking too much, but it's important to show what the leech still has in its belly in terms of biosynthesis capabilities (which genes are present and which are absent).

>> We do not understand the confusion here. the paragraph explicitly stated that a **mitochondrial** genome of the host was assembled.

"As a result of whole-genome sequencing of the novel *Haementeria* sp. COZEM-ANN-HIR-0001/002 (hereafter *Haementeria* sp.), a **full mitochondrial genome was assembled.**"

As the reviewer might not be aware of, sequencing performed on DNA samples from bacteriomes, or any other host tissue, usually leads to a number of reads originating

from the host's mitochondrion. We typically use these genomes for host taxonomic and phylogenetic inferences. This is what is referred to in this part of the text, as already expressed. In summary, no, we do not have a genome for the leech host nor a biochemical analysis of the gut content (pre- and post-feeding) or ingested blood that could aid in the metabolic inferences of host-symbiont complementation. <<

L224-224: What is the reason for this long branch characteristic (please complete with "due to..." to finish the sentence)?

>> This has been expanded to "The phylogenetic reconstruction also evidenced a long branch leading to the novel *Pluralibacter* symbiont, a feature of many long-term vertically-transmitted endosymbiont lineages **which reflects an accelerated evolution as a result of clonality and genetic drift**" <<

L273: *Provdencia* is misspelled. Please, reread the manuscript to check spelling everywhere.

>> Thank you for pointing this out. Despite our best efforts at spotting every word misspelling, some do escape. We also caught a couple extra "*Haementeria*". <<

The conclusion is well balanced.

>> Thank you. <<

Overall, the results are of good quality, even if they don't add anything new to the field of nutritional symbiosis. They do, however, draw attention to a very promising model that should be exploited with an Evo-Devo approach.

>> Thank you. We believe the work does not necessarily represent one that will significantly move the field (very few in history have), but it does add the novel knowledge of a *Pluralibacter* lineage evolving as a nutritional symbiont in *Haementeria*, its metabolic capacities, and the full mitochondrial genome of a novel member of the leech genus. We agree that *Haementeria* and other bacteriome-bearing leech genera, are great systems to be further studied from not only an Evo-Devo approach. <<

Reviewer 2:

The manuscript by Sosa-Jiménez et al., entitled "A new guest to the blood feast: a novel 2 symbiotic lineage associated with a haematophagous leech from the genus *Haementeria*" is an interesting example of how endosymbiosis has allowed insects to adapt to nutrient-deficient diets with the help of bacteria from different clades, which undergo a convergent evolution to be able to provide the same essential products to related hosts feeding on similar diets (blood in this case) in different planet regions.

I find the topic very interesting, and I have no major concerns about the manuscript as a whole, but I would like to make some comments that might help to clarify or improve some parts of the text.

The Introduction contains an excess of references which are not always relevant or the most appropriate. It might be interesting to include some previous reviews instead, since all work on the subject cannot be cited.

>> We agree on the general view of the reviewer that reviews should be cited, though when appropriate. Our general view is to cite a review when referring to something it exposes a novel integrated insight from a topic, and not only when it mentions it as part of the literature review. In the latter case, the primary articles should be published, as they are the ones the citation is actually intended to refer to. Otherwise, this leads to two grand problems of scientific literature: chain citations and citation hoarding from reviews. When referring to a topic, we have taken care of citing sufficient publications that span diversity (and thus be able to make general statements) within what is stated. For example, "Species in the Glossiphoniidae family are proboscis-bearing leeches, whose blood-feeding members have evolved different bacteriome morphologies to host endosymbiotic bacteria, namely belonging to the alpha- and gammaproteobacteria (Kikuchi and Fukatsu, 2002; Manglicmot et al., 2020; Perkins et al., 2005; Siddall et al., 2004)". These references include the diversity of such associations in leeches, as no up-to-date review is available within this topic. Another example, "Often, these microorganisms have been found to inhabit intracellularly in specialised host cells referred to as bacteriocytes, which together make up so-called bacteriomes (Boyd et al., 2016; Kikuchi and Fukatsu, 2002; Manglicmot et al., 2020; Perkins et al., 2005)". These references span a variety of bacteriome structures across different organisms. Again, no up-to-date review on the topic is available. Finally, the manuscript includes only 20 references, in its current stage. We would hardly judge this number as being excessive.
<<

L34: Considering that the hypothesis of endosymbiont replacement is the main conclusion in this manuscript, the presentation of this evolutionary outcome should be more extensively presented in the introduction for those readers that would be not so familiar with the topic.

>> We appreciate this comment and have now expanded on this topic in the introduction. <<

L122: "Surviving" is a rare Word election to talk about retained contigs.

>> We have changed to "Remaining". <<

L123: *Helobdella robusta* appears only once here in the whole text. I guess the authors use this leech of the family Glossiphoniidae because its genome has been sequenced, but they should explain so for the readers. It might be interesting to talk about it in the introduction.

>> We agree that for readers unfamiliar with leeches, no introduction of this taxon would not be adequate. Thus, we have added the following explanation following the mentioning of *H. robusta*: "*H. robusta* was selected for binning of leech-derived contigs given that, from the currently available leech genomes, it is the one originating from the closest relative to *Haementeria* spp." <<

L134: The final number of contigs corresponding to the endosymbiont genome are not clear. The authors mention they are 10 contigs with overlapping ends (line 134) or 7 partially-overlapping contigs (line 196). Why the circular structure could not be resolved? Couldn't that be solved with long reads with PacBio or Nanopores, or even using directed PCR and Sanger sequencing?

>> We agree that we did not mention that the re-mapping (to the 10 binned contigs assigned to the endosymbiont) and re-assembly resulted in the 10 contigs "becoming" 7 partially overlapping ones. We certainly made a small mistake in the second mentioning of 10 contigs, as it should have read 7. A good workflow for these sort of metagenomic assemblies should always include a step or remapping reads to binned contigs followed by re-assembly of the mapped reads, as this improves the assembly (due to really all assemblers having assumptions closer to those of a single genome). We have tried to make this clearer now in the methods when mentioning the 10 binned contigs, and correcting the mention to the 7, as follows: "The aforementioned binned contigs were used for mapping back the reads from *Haementeria* sp. **to the two bins (endosymbiont and mitochondrion)** using Bowtie2 v2.5.0 (Langmead and Salzberg, 2012). This was followed by re-assembly of the mapped reads using SPAdes as described above. This resulted in a single mitochondrial contig and 7 contigs (with overlapping ends) belonging to the putative bacterial endosymbiont." <<

L159: Please review sentence "We then those consisting..."

>> It has been corrected, seems a word was missing. It now reads "We then **subset** those consisting exclusively of single-copy core genes" <<

L215: It would be interesting to provide a phylogenetic reconstruction showing the distance between the two endosymbiotic bacteria found in the family Glossiphoniidae. Is there a single *Providencia siddallii* strain previously described or there are different strains in different related insects?

>> While we agree this would illustrate both symbionts in one figure, such a phylogenetic reconstruction would imply inferring a phylogeny almost to the order level (Enterobacteriales; see Fig. 8 from Adeolu *et. al.* 2016. *IJSME*, 66(12):5575-5599; 10.1099/ijsem.0.001485), including the *Morganellaceae* (to which *Providencia* belongs to), *Hafniaceae*, *Yersinaceae*, *Pectobacteraceae*, *Erwinaceae*, and *Enterobacteriaceae*. Because of this, we refrain from this and only provide a phylogeny for the *Enterobacteriaceae* using *Erwinaceae* as an outgroup (full phylogeny in new fig. S1, old fig. S2). Relating to the *P. siddallii* strains found in leeches, they are a monophyletic group with a clear single origin (and thus, treated as a single species). Currently, three strains (i.e. from three different *Haementeria* hosts) are sequenced (see table 1 for genome characteristics and 10.1093/gbe/evad164 for a publication analysing all hitherto sequenced strains). <<

Table 1: The amount of rRNA genes is calculated based on the coverage of the rRNA operon contig. Does that mean that this is one of the reasons to be unable to close the genome?

>> Yes, as explained in the text, the coverage of the rRNA gene operon contig was used to calculate the number of copies: coverage was triple to that of the rest of the large contigs (i.e. the non-repetitive ones). Thus, this is a reason why we were not able to close 3 out of 8 gaps, but not the remaining ones. The others land in different repeated sequences. <<

L273: Correct "Providencia"

>> Corrected. <<

References: Please check. I detected that reference 9 and 77 are the same.

>> Yes, Indeed. This was definitely a problem introduced at some point when inserting the references in bib format, as the paper was initially available in 2016 and published in 2017, we ended up generating two bib entries for the same item (despite only citing the correct one in the text). We have corrected this, and thus, references are now correct and without non-cited items. <<

Re: Spectrum04286-23R1 (Discovery of a novel symbiotic lineage associated with a haematophagous leech from the genus *Haementeria*)

Dear Dr. Alejandro Manzano-Marín:

Your manuscript has been accepted, and I am forwarding it to the ASM production staff for publication. Your paper will first be checked to make sure all elements meet the technical requirements. ASM staff will contact you if anything needs to be revised before copyediting and production can begin. Otherwise, you will be notified when your proofs are ready to be viewed.

I detected a mislabeling on Figure 1. The title (in bold) before the explanations should be "Phylogenetic tree displaying relationships among *Haementeria* species, using *Placobdella* spp. and *Helobdella* spp. as outgroups." instead of "Phylogenetic placement of the novel *Haementeria* sp." (which is the title of Figure 2 in this modified version of the manuscript), but the ASM production staff can manage to change it.

Sincerely,
Rosario Gil
Editor
Microbiology Spectrum